# Technical note: Low meteorological influence found in 2019 Amazonia fires

Douglas I. Kelley[1]*, Chantelle Burton[2], Chris Huntingford[1], Megan A. J. Brown[1,3], Rhys Whitley[4], Ning Dong[5]

[1] UK Centre for Ecology & Hydrology, Wallingford, Oxfordshire, OX10 8BB, U.K.
[2] Met Office Hadley Centre, Fitzroy Road, Exeter. EX1 3PB
[3] School of Physical Sciences, The Open University, Walton Hall, Milton Keynes, MK7 6AA, UK
[4] Natural Perils Pricing, Commercial and Consumer Portfolio and Product, Suncorp Group, Sydney, Australia
[5] Department of Biological Sciences, Macquarie University, North Ryde, NSW 2109, Australia

*Correspondence to*: Douglas I. Kelley (douglas.i.kelley@gmail.com)

**Abstract** The sudden increase in Amazon fires early in the 2019 fire season made global headlines. While it has been heavily speculated that the fires were caused by deliberate human ignitions or human-induced landscape changes, there have also been suggestions that meteorological conditions could have played a role. Here, we ask two questions: were the 2019 fires in the Amazon unprecedented in the historical record?; and did the

meteorological conditions contribute to the increased burning? To answer this, we take advantage of a recently developed modelling framework which optimises a simple fire model against observations of burnt area, and whose outputs are described as probability densities. This allowed us to test the probability of the 2019 fire season occurring due to meteorological conditions alone. The observations show that the burnt area was higher than in previous years in regions where there is already substantial deforestation activity in the Amazon. 11% of

the area recorded the highest early season (June-August) burnt area since the start of our observational record, with areas in Brazil's central arc of deforestation recording the highest ever monthly burnt area in August. However, areas outside of the regions of widespread deforestation show less burnt area than the historical average and the optimised model shows that this low burned area would have extended over much of the eastern Amazon region, including in Brazil's central arc of deforestation with high fire occurrence in 2019. We show

that there is a 9% chance of the observed August fires being caused by meteorological conditions alone, decreasing to 6-7% along the agricultural-humid forest interface in Brazil's central states, and 8% in Paraguay and Bolivia dry-forests. Our results suggest that changes in land use, cover or management are the likely drivers of the substantial increase in the 2019 early fire season burnt area, especially in Brazil. Burnt area for September in the arc of deforestation had a 14-26% probability of being caused by meteorological conditions, potentially

coinciding with a shift in fire-related policy from South American governments.

**1 Introduction**

South American fires made global headlines in August 2019, with the largest increase in fire activity seen in nearly ten years (INPE, 2019; Lizundia-Loiola et al., 2020). Of the roughly 100,000 fires burning by the end of the month, around half were in the Amazon rainforest region (Andrade, 2019; INPE, 2019). While fires in drier savannah regions of South America such as the Cerrado are more common, fires in the rainforest are not a natural occurrence and are rarely ignited without human intervention (Aldersley et al., 2011). As such, fires in humid, tropical regions where the vegetation is not adapted to frequent burning (Kelley, 2014; Staver et al., 2020; Zeppel et al., 2015), have much higher tree mortality rates (Brando et al., 2014; Cochrane and Schulze, 1999; Pellegrini et al., 2017). As a result, an estimated 906,000 hectares of the Amazon biome was lost to fires in 2019 (Butler, 2017). The amount of carbon and trace gas emission was also a major concern given the high biomass of the areas being burnt, and smoke from these fires reached cities as far as São Paulo more than 2,700 km away (Lovejoy and Nobre, 2019). Usually, small-scale fires in Amazonia are associated with deliberate but localised deforestation, although in dry years, there is more risk of these fires escaping into much larger areas (Aragão et al., 2018). Hence the substantial increase in fires in 2019 sparked much debate about whether the level of burning was unprecedented, whether increased burning was driven by a drier than normal fire season and if raised levels of direct deforestation played a role (Arruda et al., 2019; Escobar, 2019).

The Amazon has not been the only place with recent unusual and high fire activity, with large-scale fire events worldwide in the last couple of years including in the Arctic, Mediterranean, Australia, UK and the US. In November 2018 over 80 people were killed in the Camp Fire in Paradise California, the most destructive in California's history, with the Camp, Woolsey and Carr Fires together costing an estimated $27 billion in damages (Nauslar et al., 2019). California 2020 fire season is set to be even more damaging (Anon, 2020), though the true extent and costs are still increasing at the time of writing. Hundreds of fires burnt in Siberia and Alaska throughout the 2019 and 2020 northern hemisphere summers, releasing over 150 Mega tonnes of $CO_2$ in 2019 abd 244 Mega tonnes in 2020 into the atmosphere (Witze, 2020). Of major concern was the release of large quantities of black carbon with the potential to further accelerate local arctic ice melt (Patel, 2019). The UK saw some burning, including a peatland fire in north-east Sutherland that doubled Scotland's carbon emission for six days in May 2019 (Wiltshire et al., 2019). Between September 2019 and February 2020, fires across southeastern Australian burnt around 18.6 million hectares, destroyed over 5,900 buildings, and killed at least 34 people (Boer et al., 2020; RFS, 2019; Sanderson and Fisher, 2020). Unusual fire events such as these are expected to increase in frequency in the future from both changes in climate and socio-economic pressures on the landscape (Fonseca et al., 2019; Jones et al., 2020). Given the concerns raised and the extent to which much of these fire events captured the attention of the public and press in recent months, in the aftermath, it is important to look at these events objectively. In particular, it is essential to determine if they were unusual in the context of the historical record and if so, what might be new and emerging drivers.

There are many ways to assess drivers of historical fire events. Some studies simply correlate individual drivers with burnt area in isolation (Andela et al., 2017; Van Der Werf et al., 2008). However, these do not consider the

complex interaction of multiple drivers on fire and are therefore unable to go beyond a loose attribution of a particular forcing to fire, which can easily be confused as causations due to the number of drivers. Fire Danger Indices (FDIs), which can capture simultaneous drivers, are useful for calculating the level of risk of a fire

spreading and becoming severe in a particular area (de Groot et al., 2015). FDIs have been adapted to assess recent and future trends in climate on fire weather (Burton et al., 2018; Jolly et al., 2015) and attribute increases in fire risk to anthropogenic changes in climate (van Oldenborgh et al., 2020). These metrics thereby provide rapid policy-relevant information for fire management (De Groot et al., 2010; Perry et al., 2020). However, FDIs by themselves do not account for fuel or ignitions, differentiating them from fire observations such as burnt

area, making them an unsuitable tool for assessing fire in the holistic context of weather, fuel dynamics, ignitions and human land and fire management (Kelley and Harrison, 2014). Fire-enabled Land Surface Models (LSMs) can, however, account for these drivers (Kelley and Harrison, 2014; Lasslop et al., 2016; Prentice et al., 2011b) to simulate a physical, observable measure of fire regimes, such as burnt area or number of fires. Some LSM fire schemes achieve this by modelling fuel moisture using FDIs (Lenihan et al., 1998; Rabin et al., 2017;

Venevsky et al., 2002). However, most LSMs have been developed to study long-term, often decadal timescale carbon dynamics and Earth System feedbacks and therefore often fail to reproduce year-to-year patterns of fire with the required accuracy to determine the causes of individual fire seasons (Andela et al., 2017; Hantson et al., 2016, 2020). Embedding fire within a complex vegetation model system also prevents rapid observation-model fusion, as iterative optimisation techniques are too computationally expensive and instabilities arise from

non-linear responses of fire to simulated vegetation and fuel dynamics. Many large scale vegetation-modelling projects, therefore, simulate up to a "present-day" that can be several months or years out of date (Friedlingstein et al., 2019; Hantson et al., 2020). This lack of predictive capability has led to calls for simulation frameworks that fuse statistical representations of fire drivers with modelling techniques that consider such interactions (Fisher and Koven, 2020; Forkel et al., 2017; Krawchuk and Moritz, 2014; Sanderson and Fisher, 2020;

Tollefson, 2018; Williams and Abatzoglou, 2016).

Kelley et al. (2019) developed a methodology which addresses this gap by coupling the same representation of processes found in simple fire enabled LSMs (Rabin et al., 2017) using a Bayesian inference framework. The main advantage of this system is that it can assess the contribution of different fire drivers directly from observations and track uncertainty in the model parameters and the models' ability to reproduce observations.

We apply this methodology here, using monthly meteorological conditions and burnt area (BA) observations to constrain and drive the model, thus capturing interannual variability within the context of changing meteorological conditions. We use this framework to answer the specific question: Did the meteorological conditions contribute to the Amazonia Fires of 2019?

## 2 Methods

The modelling protocol and optimisation framework largely follow Kelley et al. (2019), where a more detailed description may be found. Here, we summarise and outline further refinements. Monthly burnt area (BA) is modelled as a product of limitations imposed by four controls: 1) fuel availability; 2) moisture in live and dead

fuel; 3) anthropogenic and natural ignition; 4) both active suppression and landscape fragmentation effects from human land use (Table S1 in Supplement). Each control is calculated as a linear combination of its respective drivers. The impact each control has on fire is represented by a logistic curve describing the maximum allowed burnt area based on that control, and the product of all four limitations is used to determine BA.

We made several changes to the previous modelling protocol in order to utilise near-real-time meteorological and fire variables so that we can produce relevant results that closely follow the fire event, and to better describe the conditional probability distribution between parameter samples and burnt area observations. We used the MODIS Collection 6 MCD64A1 burned area product (Giglio et al., 2018) as our target dataset and replaced actual over potential evapotranspiration in the moisture control with soil moisture (Table S1). We also used both the top 10cm and 10-200cm soil moisture (Kalnay et al., 1996) as independent moisture drivers in order to capture the impact of previous drought years on deepwater availability for live fuel. As near-real-time wetday information is also not available, we replaced wet days in the calculation of dead fuel drying potential (Kelley et al., 2014) with a proxy for wetdays (*WD*), using GPCP precipitation (Adler et al., 2003) (*pr*) based on (Prentice et al., 2011a):

$$WD = 1 - e^{-wd \, x \, pr} \qquad\qquad (1)$$

where *wd* is an optimised parameter.

All variables were resampled and, where necessary, interpolated to a monthly time-step as per Kelley et al. (2019). All driving variables were provided on a resolution of 2.5° except land use, provided at 0.5°. We, therefore, choose to regrid all datasets to a resolution of 2.5°, as interpolating to a finer resolution would provide no new information about the meteorological drivers tested. MCD64A1, soil moisture and equilibrium fuel moisture content were processed using the "rgdal" (Bivand et al., 2016) and "raster" (Hijmans and van Etten, 2014) packages in R (R Core Team, 2015). For MODIS Vegetation Continuous Fields (VCF) fractional covers (Dimiceli et al., 2015), tiles were merged and resampled to the model grid using the "gdal" package (GDAL/OGR contributors, 2018). Land use, population density, precipitation, humidity, temperature and lightning, were processed using the iris package (Met Office, 2013) with Python version 3 (Python Software Foundation, https://www.python.org/).

The model was optimised against MCD64A1 burned area (Giglio et al., 2018) for the period July 2002 to June 2018, which was the common period for all datasets (Table S2) over South America, south of 13°N. We used the same Bayesian Inference technique as per Kelley et al (2019). Bayes' theorem states that the likelihood of the values of the set, ß, which contain our 24 unknown parameters (i.e. the 21 parameters from Kelley et al (2019), *wd* from equation 2, and error term parameters $P_0$ σ in equation 3) and our known model inputs, given a set of observations $Y_s$ is proportional to the prior probability distribution of ß ($P(ß)$) multiplied by the conditional probability of $Y_s$ given ß:

$$P(\beta|Y_s) \ \propto \ P(\beta) \ \cdot \ P(Y_s|\beta) \tag{2}$$

As 41.47% of the burnt area observations are zero, and the remaining are normally distributed under *logit* transformation (Fig. A1). We, therefore, defined the likelihood, $P(Y_s|\text{ß})$, using a zero-inflated normal distribution on the logit transformed burnt area, as opposed to a simple normal distribution as used in Kelley et al (2019). This better described the observational to the simulated difference in burnt area during times of very low or very high burning. Our zero-inflation likelihood term is therefore described as:

$$P(Y_s = 0|\text{ß}) \ = \ 1 \ - \ BA_i^2 \ \times (1 \ - \ P_0)$$

$$P(Y_s > 0|\text{ß}) \ = \ [1 \ - P(Y_s = 0|\beta)] \ \times \ \frac{N}{\sigma\sqrt{2\pi}} exp\left\{ \Sigma_i^N \left( \frac{logit(y_i) - logit(BA_i)}{\sigma} \right)^2 \right\} \tag{3}$$

where $i$ represents an individual data point, $y_i$ is the burnt area observations, $N$ is the observation sample size and $logit(x) = log\left(\frac{x}{1-x}\right)$.

The posterior solution was inferred for all model parameters using a Metropolis-Hastings Markov Chain Monte Carlo (MCMC) step with the PyMC3 Python package (Salvatier et al., 2016), running 10 chains each over 10,000 iterations. We used all of the 44750 grid cells on our 2.5° grid and monthly time step for 16 years in our assimilation procedure. This is a departure from Kelley et al. (2019), where only 10% of grid cells were used, as our sample size was much smaller and we did not face the same computational demand. Due to our sample size, our likelihood dominates over our priors, and as with Kelley et al. (2019), priors predominantly were employed to set physically plausible bounds on our parameters.

Once optimised, the model was then run from January 2002 - December 2019 and so the trained model was in a predictive mode for 2019. Due to data availability at the time of writing, July 2017 - June 2018 land cover, land use and population density were recycled for July 2018 onwards (Table S2). We sampled 100 parameter ensemble members from the last 5000 iterations of each of the 10 chains, providing us with 1000 ensemble members to estimate the models' posterior solution to equation 2. Sampling was performed using the iris package (Met Office, 2013) with Python version 3 (Python Software Foundation, https://www.python.org/). The posterior solution provides an estimate of the burnt area based on the parameter uncertainty of our model, corresponding to the yellow areas in time series in Fig. 1 and 2. The mean burnt area for a particular parameter combination ($\overline{BA_\text{ß}}$), was obtained from:

$$\overline{BA_\text{ß}} \ = \ \int_0^1 P(BA|\text{ß}) \times BA \ dBA \tag{4}$$

$\overline{BA_\text{ß}}$ was evaluated using the implementation of the fireMIP benchmarking metrics (Hantson et al., 2020; Kelley et al., 2013; Rabin et al., 2017) as per Kelley et al. (2019). We also performed additional benchmarking metrics of the models' ability to reproduce seasonality and inter-annual variability of fire. See Fig. S1-S3 and model evaluation supplement for validation methods and results.

We chose five regions (marked A-E in Fig. 1, 2. See Fig. A2 for locations) to represent forest areas already under pressure from deforestation. Regions A-C form a transect (west to east) across the agricultural-humid tropical forest interface in Brazil's arc of deforestation, often associated with deforestation (Fig. S4), whereas D and E regions are found in agricultural regions embedded in savanna and grassland regions that experience frequent burning:

A. Acre, Southern Amazonas States and Brazilian/Peruvian border
B. Rondônia and Northern Mato Grosso, Brazil.
C. Tocantins, Brazil,
D. Maranhão and Piau in coastal deforestation regions
E. Brazilian, Bolivian and Paraguayan border

We also assessed an overall Area of Active Deforestation (AAD) in the Amazon region (Fig. A2). This area is defined as the parts of South American southern tropics with significant decreasing tree cover trends, as seen in VCF (Dimiceli et al., 2015) and significant increasing agricultural fractions in the HYDEv3.1 dataset (Klein Goldewijk et al., 2010). Trend analysis used the same technique described in Kelley et al. (2019), where we took significance as $p<0.05$ on the linear trend for each month in the year on logit transformed cover variables, using the greenbrown R package (Forkel and Wutzler, 2015). AAD was additionally assessed over 3 sub-regions areas, primarily to evaluate the models' historic performance and assess the increase in 2019 fires across the humidity gradient:

F. Area of Active Deforestation
G. The Southern end of agricultural-**humid tropical forest** interface in Brazil's central states, often associated with the arc of deforestation in Brazil's central states
H. **Drier savanna and woodland** in Cerrado and Caatinga in the eastern Basin were land has already been heavily converted to agriculture
I. Southern **seasonally dry-deciduous** Chiquitano and Gran Chaco forests, mainly along the Amazon, La Plata watersheds.

We assessed the probability of 2019 fire activity being explained by information provided to the model in three ways (Table 1):

1. The **likelihood of observed monthly burnt area** based on the information provided to our model. In the predictive model, the probability of a burnt area $y$ (where $y$ can be outside training data $Y_s$, as is the case for our year 2019 analysis) being explained by our model ( $Pred(y)$ - full model uncertainty, or model error, in tan areas on time series in Fig. 1, 2) is proportional to the probability of $y$ given a parameter set, $\beta$, weighted by $P(\beta|Y_s)$ :

$$Pred(y) \propto \int_{PS} P(\beta|Y_s) \times P(y|\beta) \, d\beta \qquad (5)$$

Where the observed burnt area, $y$, falls within the model's full posterior ($L(y)$) is then the sum of all probabilities greater than $y$,

$$D(y) = \int_y^1 Pred(y) \, dBA \tag{6}$$

$D(y) \sim 0, 1$ therefore suggests y is towards the extremes of the posterior. As our posterior solution is not normally distributed, observations can fall at the extremes (i.e. when there is no burnt area, $y = 0$ and by definition $D(y) = 1$), and still have a high likelihood (i.e. if $P(Y_s = 0| ß)$ in equation (3) is much greater than 0. See Fig B2 as an example). We, therefore, define the significance of $D(y)$ as the probability of $y$ occurring by chance ($pv(y)$) from the sum of all probabilities below $P(y)$ (Fig. A2):

$$pv(y) = 1 - \sum_{i \in 2Z}^{i \le n} \left[ \int_{y_{i-1}}^{y_i} P(x) \, dx - P(y) \times (y_i - y_{i-1}) \right] \tag{7}$$

where $\{y_1, ...., y_n\}$ is the set of solutions to $P(y_i) = P(y)$

Whenever $D(y)$ and $pv(y)$ are low indicates burning significantly higher than expected by the model in that month.

2. The **likelihood burnt area would have been higher than the annual average**, i.e. the fraction of the model's full posterior greater than the model's annual average climatological posterior (the point where the vertical lines cross 1 in right-hand columns in Fig. 1,2). A climatological burnt area *clim* for a given month, $m$, in the year (i.e. January, February, etc.) can be calculated from the convolution of each year's posterior solutions, $\beta_{yr, m}$. Note that it is the model inputs, incorporated in $\beta_m$, and not the model parameters that vary with time:

$$P(BA \mid clim_m) = P\left(BA \mid \beta_{2001, m} \cap \beta_{2002, m} \cdots \cap \beta_{2019, m}\right)$$

Where $P\left(BA/2 \mid \beta_i \cap \beta_j\right) = \int_0^{BA} P\left(BA - x \mid \beta_j\right) \times P(BA \mid \beta_i) \, dx$ and

$$P\left(BA/3 \mid \beta_i \cap \beta_j \cap \beta_k\right) = \int_0^{BA} P(BA - x \mid \beta_k) \times P\left(x/2 \mid \beta_i \cap \beta_j\right) dx \tag{8}$$

The probability of an anomaly $A$ in a given year, $yr$, for month $m$ is, therefore:

$$P\left(A \mid \beta_{yr, m} \cap clim_m\right) = \int_0^1 P(A \times BA \mid \beta_{yr, m}) \times P(BA \mid clim) \, dBA \tag{9}$$

The likelihood of a year having a higher anomalous, $A$, is the sum of probabilities of $< A$ :

$$L\left(A|\, \beta_{yr,\, m} \,\cap\, clim_m\right) \;=\; \int_0^A P\left(A|\, \beta_{yr,\, m} \,\cap\, clim_m\right) dA \tag{10}$$

And the likelihood of the year an average burnt area is given $L\left(1|\, \beta_{yr,\, m} \,\cap\, clim_m\right)$

3. The **likelihood of the observed anomalous year** occurring is given by :

$$L\left(\tfrac{y_{m,\, yr}}{\overline{y_m}} \,/|\, \beta_{yr,\, m} \,\cap\, clim_m\right) \tag{11}$$

where $y_{m,\, yr}$ is the burnt area for the month, $m$, and year, $yr$, in question, and $\overline{y_m}$ is the climatological average of that month

## 3 Results

### 3.1 Burnt area from 2001-2018

The highest historic burnt areas are found in the Savanna regions of tropical South America (Fig. 3), though some burning still occurs in forested areas, particularly in areas which have experienced an increase in agriculture and decrease in tree cover since 2002 (Fig. S4). The model reproduces this spatial pattern, and the models full posterior encompass the full range of burnt areas (Fig. 3, and benchmarking SI). Burnt area starts to increase in May and dies out in October throughout most of the Area of Active Deforestation, though can start as late as July in more humid areas can continue through to December in drier Savanna (Fig S3). The bulk of the burnt area occurs in August and September. September typically sees the highest burnt area in central Brazil, whereas fire peaks in August around Bolivia and Paraguay (Fig. 2F, S3). Our model reproduces this seasonal pattern in burning across all regions (see benchmarking SI), including onset and peak (Fig S3). As our model maintains constant human ignitions and suppression throughout the year, this suggests that the seasonal pattern can be largely reproduced from meteorological variations. Though a slight increase in uncertainty in early fire season burning could point to increased human ignitions not captured in the model (Fig. S3).

Unusually high levels of burning occurred in 2004 in the Bolivian/Paraguayan dry forest (red line in Fig. 1E), 2005 in the eastern arc of deforestation (Fig. 1A) and Paraguay dry forest (Fig. 1E), 2007 in monsoonal coastal forests (Fig. 1D) and 2010 in Bolivia and Paraguay dry forests (Fig. 1D and E). 2005 and 2010 burning have previously been associated with droughts driven by a Tropical North Atlantic warming anomaly (Marengo and Espinoza, 2016). The model reproduced the spatial pattern of this increased burning in 2005 and 2010 (Fig C1, C2). In our different regions, observed levels of buring fall within, although at the higher end, of our models posterior (Fig. B1, B2) with a high value of expectation (height of the posterior curve in Fig B1, B2) and high

p-value ( blue shaded area, B1 and B2). This, along with the model high spearman's rank performance (Fig S1 and S2) suggests that the model is able to capture the interannual variations driven by meteorological conditions. Deforestation rates in 2004/05 were high (Marengo et al., 2018), and an increase in fire activity in 2007 has also been linked to deforestation across the Amazon (Morton et al., 2008). Additionally, in the early part of our observational record, much of the region has been shown to be less coupled to meteorological drivers

and more heavily influenced by human fire and land management (Aragão et al., 2018). This is reflected by the improved performance of the model, which depends solely on changing population density and land use cover and not on changes in landscape management, during this later period in the AAD (Fig 2F, 2011 onwards), particularly in areas dominated by agriculture (Fig. 2H).

On the whole, the frameworks posterior is better able to encompass extremes in observations in humid regions

(Fig 2G vs Fig 2H, I), particularly across the Brazilian arc of deforestation (Fig1A-C vs D and E). 19 out of 204 months up to 2018 for the AAD (~9%) fall outside the 90% confidence interval (tan in Fig 2F), suggesting that the frameworks posterior accurately describes the occurrence of more extreme months for the region as a whole. That only 13 months out of (204 months x 5 regions) 1020 months (~1%)  fall outside the posterior for smaller regions (Fig 1) suggest that the posterior is wider than expected. Our assessments of mismatch between

observations and model for these regions will, therefore, likely be conservative, particularly for humid regions B and C, with no months prior to 2019 falling outside the 90% confidence interval.

**3.2 Burnt area in 2019 in context**

The year 2019 burnt area during the early fire season (defined as June to August) was higher than the 2002-2019

average in areas of recent historical deforestation, despite a lower than average burnt area over much of the rest of the continent (Fig. 3). The AAD as a whole saw the 3rd highest levels of burning in the fire season (Fig. 2F, August), behind 2007 and 2010. 11% of the AAD, particularly around the central region of Brazil's arc of deforestation, experienced more burning in August than any previous year since our observation record started in 2002 (Fig 1B, C).  Despite burnt area returning to normal levels in September across most of the AAD (Fig.

4), burning remained high in humid forest areas (Fig. G), particularly in central Brazil (Fig. B, C). Burning also remained higher than average along the border between Brazil, Bolivia and Paraguay (Fig. 1E, Fig 2I and Fig 4). This meant that, while the burnt area was higher than usual in 2019, it was not exceptionally higher over the entire fire season (June -September) for the entire AAD, though individual regions still stand out as having much higher burning than any previous year, particularly in Brazil's central states (Fig2. B and C).

**3.3 Climatic conditions in 2019**

The model shows with high confidence that most of the Eastern Amazonia should have, in fact, experienced less fire than normal for June-August when accounting for 2019 meteorological conditions. This expected low fire rate included areas in the Brazilian humid forest-agricultural interface in the AAD that saw higher than annual

average burning in observations (Fig. 1C and Fig. 3). Western Amazon shows an increase in fire compared to the annual average (Fig 3.). The observed burnt area, however, still exceeds the model in all our regions in 2019 (Fig. 1) except region D in the already heavily converted agricultural land near the Brazilian coast (Fig. S4). Only 0-2% area of the AAD showed unprecedented high burning in the model, compared to the 11% in observations. Observed burnt area in August falls outside the full models posterior (at 90% confidence interval) for the AAD (Fig 2F), with only a 9% likelihood of being explained by the model (Table 1). This is particularly prominent in more humid areas, with a 10% likelihood in humid forests (Fig 1G, Table 1, G) - tied as lowest likelihood in the observational record with September 2005, compared to 17% likelihood in seasonally dry forests regions (Fig. 1I) and 17% the driest, savanna areas (Fig. 1H) where observations tend to fall outside the model posterior more often (see 3.1). Regions B, C in Central Brazil, and E on the Brazilian, Bolivia and Paraguay border are even less likely to be explained by the model (7% for B, 6% for C and 8% for E, Table 1), despite all previous months falling within the full model's posterior confidence range in these regions, except for August and September in 2005 in region A. Although it is more likely that burnt area regions A and D at either end of Brazil's arc of deforestation could be explained by the model (18% and 20% respectively).

The observed anomaly for August 2019 is higher than the model across all regions except D. This is particularly prominent in regions B and C, where observations show that burnt area was 196% and 138% greater than the August average (Table 1). Whereas the model suggests that meteorological conditions alone should have resulted in a fire season with a 16-22% increase (based on 5-95% parameter uncertainty range for parameter uncertainty) in burnt area in B and 2% reduction to 4% increase for C compared to the August average, with only a 57% and 53% chance of a greater burnt area than the average for B and C respectively. The likely occurrence of the observed anomaly was 7 and 10% for B and C, respectively (Table 1) - much greater than any previous year (Fig 1B and C, August column).

The higher observed anomaly vs the model extends over much of the AAD (Fig. 2 "August" column, red points). The model suggests a 4-6% reduction for the AAD, with a 49% probability of greater than the annual average burnt area (Table 1). By comparison, the observed burnt area was 45% greater than the annual average, with a 20% likelihood. Again, the observed anomaly seems to be least likely in more humid regions. For our humid area, G, the model suggests a small (10-14%) increase in burnt area, with only a 12% probability of 107% increase seen in observations, whereas the 5% observed reduction in drier savanna regions in H seems to be in line with the model (at 67% likelihood).

By September, there was less disparity between observations and models. In region B and C, for example, observations had a 14 and 26 % probability of being explained by the model, and a 66% and 25% likelihood of the anomalous year as seen in the observations. Across the AAD, the more modest 16% increase in the observations had a likelihood of 40%.

**4 Discussion**

The observed spatial pattern of burnt area in June-August 2019 shows that unprecedented burning was only seen in Brazilian regions normally associated with deforestation. Our modelling framework demonstrates that, based on meteorological conditions alone, reduced burning seen across the rest of Tropical South America should have extended into these regions. Specifically, our analysis suggests that there is only a 9% probability that the levels of burning in the early fire season would have been caused by 2019 meteorological conditions or natural ignitions alone (time series Fig. 1, 2). Eastern areas normally associated with deforestation did show expected levels of burning, but in the western and central parts of the arc of deforestation and Bolivia and Paraguay dry forests, burning was much higher. Here there is a 6-8% of such high levels of burning compared to the background rate (Fig. 1 "August" column), with areas where agriculture meets more humid forest seeing the most unusual levels of burning. As our model's posterior reflects the levels of burning in previous dry years, we can eliminate drier conditions as a possible driver of increased 2019 fires. We also account for and therefore eliminate, longer-term drier conditions through deep soil moisture as a possible driver. The cause of increased burning in 2019 is therefore either a driver left static in the model for 2019, or a process not considered. Because of the non-availability of near-real-time data, drivers held unchanged at 2018 values for 2019 are tree cover, land use and human population. The only plausible way tree cover could have substantially changed is through increased deforestation rates (Zhang et al., 2015). Thereby changes in drivers not accounted for in 2019 would only have caused increased burning through direct human manipulation of the landscape rather than the particular meteorological features of that year.

Improved descriptions of evolving changes in human fire and landscape interactions over time may also be required to capture direct human-driven changes in burnt area. This is likely to include changes in demography or human behaviour. For example, we currently account for the impact of a changing population on fire starts and suppression, but not how fire ignitions per person change over time. An evolving policy could have also been the cause of the unusual fire activity. It should also be noted that observed fire activity returned to expected levels given meteorological conditions in September over most of the deforestation region (Fig. 1, 4). This reduction could be after the June-August fires received international media coverage, triggering efforts in combating fires from South American governments (BBC news, 2019; NASA, 2019).

**5 Conclusion**

In this study, we have used a novel Bayesian modelling approach, which tests the likelihood of observed extremes in fire against inferred historical relationships by tracking uncertainties in modelling fire in the land surface. Our framework provides a rapid assessment of whether there was any influence of meteorological conditions across the Amazon that exacerbated fire levels in 2019.

The model predicts a lower burnt area than we see in the observations for Amazonia during June-August 2019, indicating that from observed meteorological data alone, we would not expect 2019 to be a high-fire year. This

result points to socio-economic factors having a strong role in the high recorded fire activity. Specifically, we conclude that it is likely (>90% probability) based on past relationships between burnt area and meteorological conditions, that the weather conditions did not trigger the increase in burning in Amazonia during the early fire season in 2019. This result holds over the entire area of active deforestation and furthermore is likely (93%) in central Amazonia.

## Acknowledgements

The authors declare no conflict of interest. The contribution by D.K. and M.B. was supported by the UK Natural Environment Research Council through The UK Earth System Modelling Project (UKESM, grant no. NE/N017951/1). C.H. gratefully acknowledges the NERC CEH National Capability Fund. N.D. is supported by the Australia Research Council (DP170103410) and Macquarie university research seeding grant to Dr Melanie Zeppel.

GPCP Precipitation data provided by the NOAA/OAR/ESRL PSD, Boulder, Colorado, USA, from their Web site at https://www.esrl.noaa.gov/psd/. NCEP Reanalysis data provided by the NOAA/OAR/ESRL PSD, Boulder, Colorado, USA, from their Web site at https://www.esrl.noaa.gov/psd/

## Code/Data Availability

The model code and Bayesian inference framework used to support the findings of this study are archived at https://doi.org/10.5281/zenodo.3817456. Model output is archived at https://doi.org/10.5281/zenodo.3817467. Driving data availability is listed in Table S1.

## Author contributions

DK, CB, MB and CH developed the modelling framework; DK, RW designed the Bayesian inference framework; DK, MB and ND collated and regridded input data. DK, CB, CH performed the analysis. CB wrote the first draft of the paper with input from DK, CH and ND. All authors contributed to the final manuscript. The authors declare no competing interests.

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

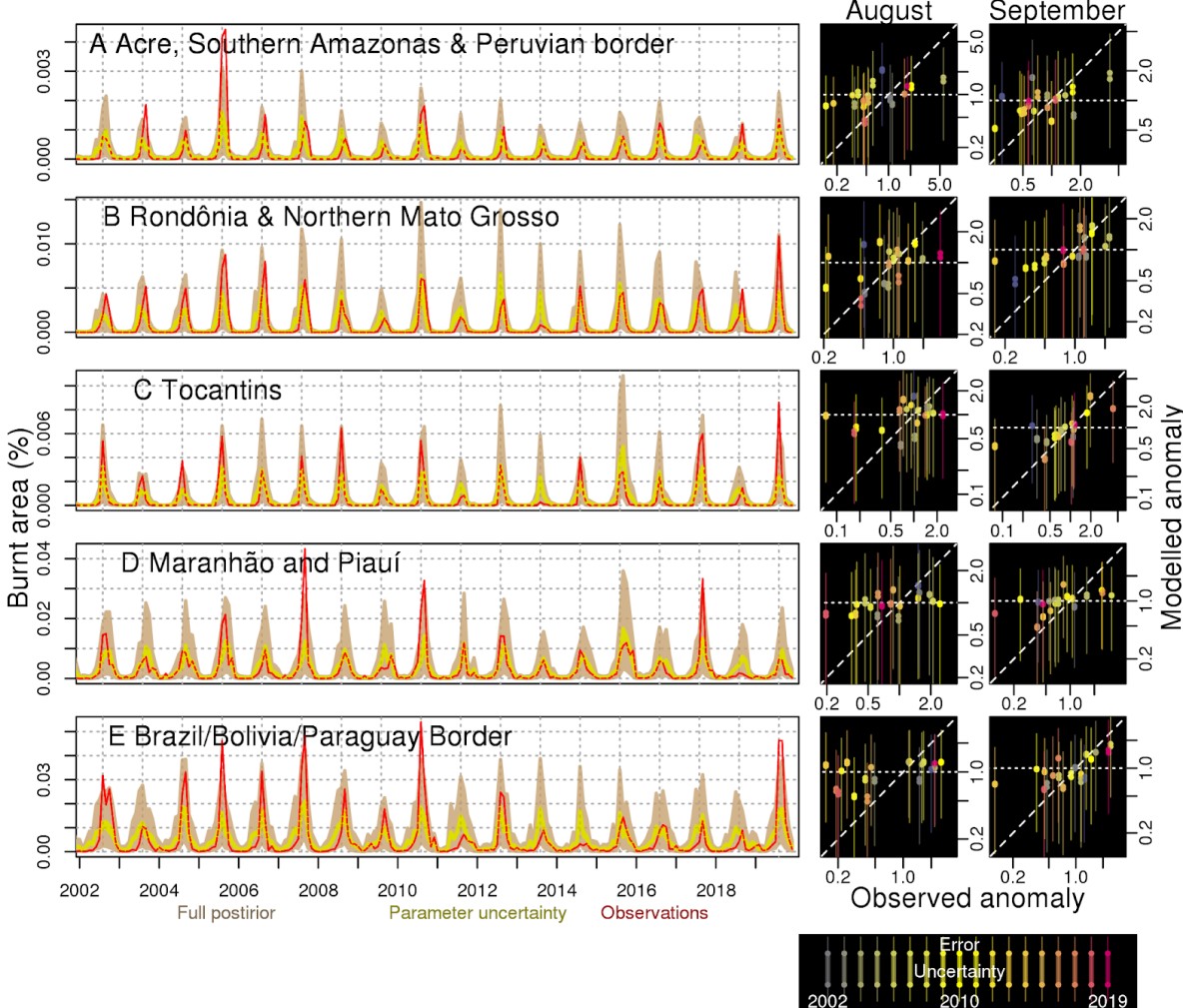

Figure 1: Time series and fire season anomalies for modelled and observed burnt area. See Fig. A2 for locations of A-E. Red lines show monthly burnt area observations from MCD64A1, yellow shows model accounting for parameter uncertainty (10-90%) and brown shows full model uncertainty (10-90%). The red line is dashed when observations and model accounting for parameter uncertainty overlap. Vertical grid lines are positioned for August each year. Right-hand plots show observed (x-axis) and modelled (y-axis) anomaly, calculated as 2019 burnt area over 2002-2019 climatological average burnt area for (first column) August and (second column) September. The colour indicates the year, with 2019 in red. Thin lines show 10-90% full model uncertainty, while dots and thick line indicate 10-90% parameter uncertainty

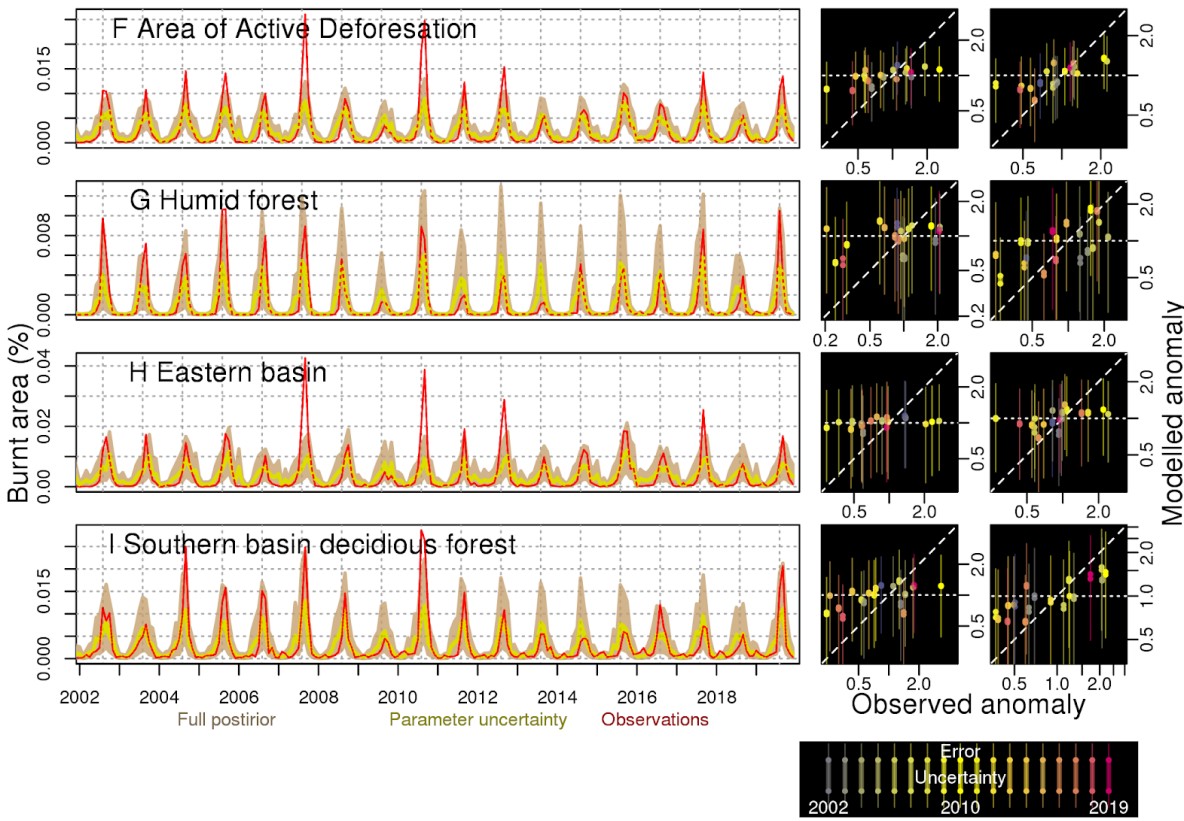

**Figure 2: As Fig. 1, but for the "Area of Active Deforestation" region which incorporates areas where there has been a significant increase in agriculture and decrease in tree cover. See Fig. S4, and regions and increased agriculture and decreased tree cover in the (G) humid tropical forest, (H) savanna and (I) dry-deciduous Forest.**

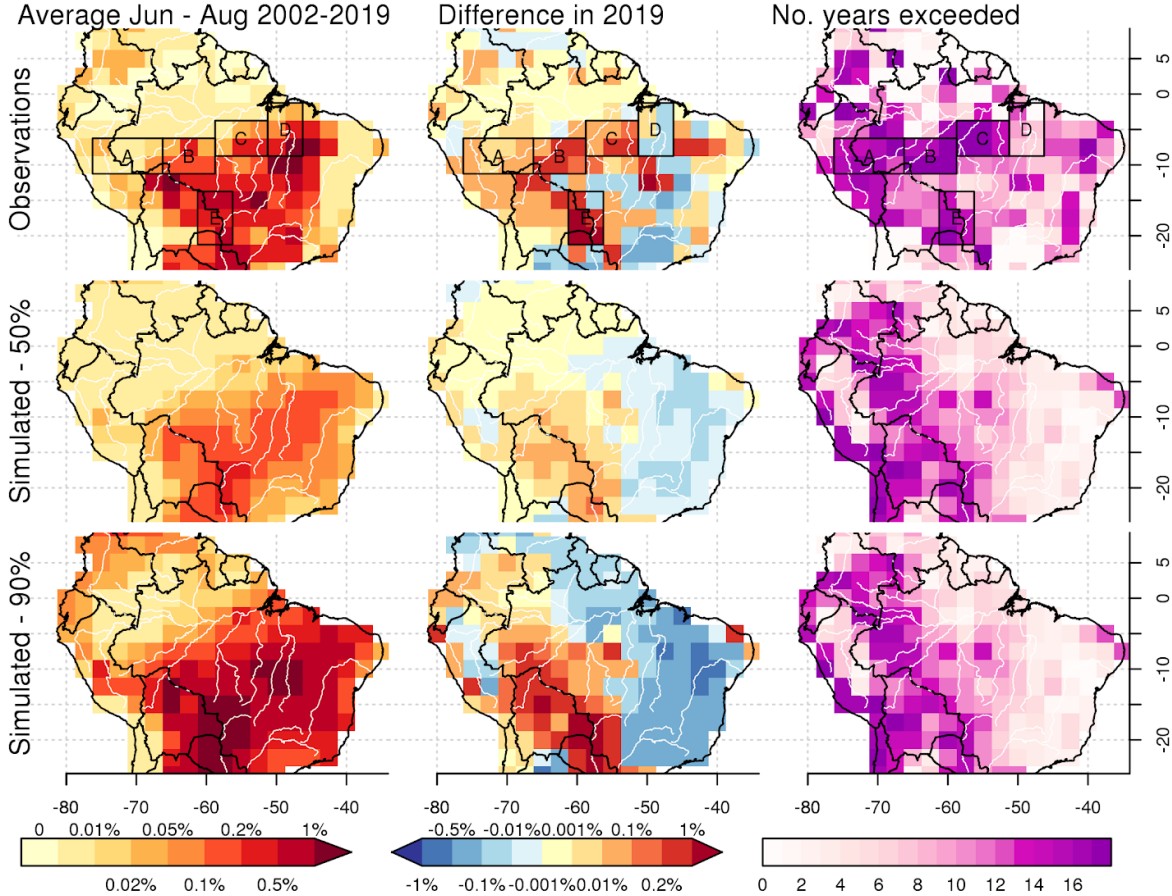

**Figure 3: Maps of modelled and observed % burnt area. First row: observed burnt area, June-August 2002-2019 annual average (left) and difference between June-August 2019 and 2002-2019 average (centre), and the number of years 2019 burnt area exceeds (right). Second and third-row: as top row, for model posteriors 5% and 95% percentile.**

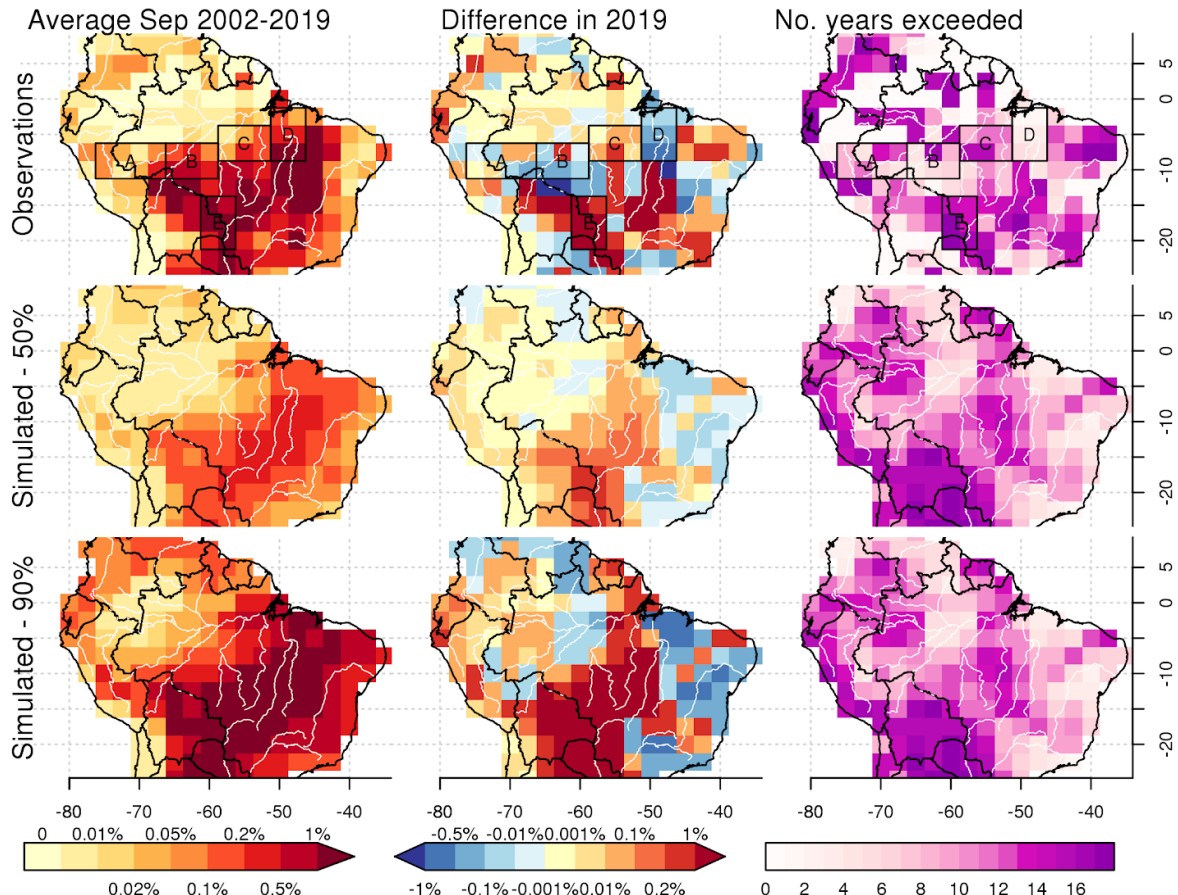

**Figure 4: Same as Fig. 3 but for September.**

Table 1: Observed and model anomaly in burnt area for August and September 2019 as a fraction of August and September averages 2002-2019 across selected regions (see methods). Red indicates more burning than normal, blue less and yellow around average burning. The model is expressed as 5-95% of the posterior accounting for parameter uncertainty. Likelihood gives the percentage probability that (1st column) the observed burnt area as suggested by the model and (2nd) it's p-value; (3rd) that the model suggests a higher than average burnt area for the given month, and (4th) that the model captures the observed anomaly based on the full model posterior.

| Regions | | Observed anomaly | Model anomaly | | Likelihood (%) | | | |
|---|---|---|---|---|---|---|---|---|
| | | | 5% | 95% | Burnt area | p-value | Higher than average | Anomaly |
| A | Aug 2019 | 1.82 | 0.88 | 0.96 | 18 | 0.1 | 52 | 18 |
| | Sep 2019 | 0.57 | 1.29 | 1.32 | 18 | 0.09 | 48 | 58 |
| | | | | | | | | |
| B | Aug 2019 | 2.96 | 1.16 | 1.22 | 7 | 0 | 57 | 7 |
| | Sep 2019 | 0.76 | 1.11 | 1.25 | 14 | 0 | 57 | 66 |
| | | | | | | | | |
| C | Aug 2019 | 2.38 | 0.98 | 1.04 | 6 | 0.07 | 53 | 10 |
| | Sep 2019 | 1.15 | 0.94 | 1.04 | 26 | 0.98 | 51 | 25 |
| | | | | | | | | |
| D | Aug 2019 | 0.68 | 0.68 | 0.7 | 20 | 0.76 | 46 | 64 |
| | Sep 2019 | 0.46 | 0.91 | 0.95 | 43 | 0.88 | 52 | 53 |
| | | | | | | | | |
| E | Aug 2019 | 2.18 | 1.09 | 1.14 | 8 | 0.03 | 54 | 14 |
| | Sep 2019 | 2.31 | 1.21 | 1.28 | 12 | 0.02 | 56 | 24 |
| | | | | | | | | |
| F | Aug 2019 | 1.45 | 0.94 | 0.96 | 9 | 0.01 | 49 | 20 |
| | Sep 2019 | 1.16 | 1.06 | 1.08 | 9 | 0.19 | 50 | 40 |
| | | | | | | | | |
| G | Aug 2019 | 2.07 | 1.1 | 1.14 | 10 | 0.01 | 55 | 12 |
| | Sep 2019 | 0.74 | 1.09 | 1.15 | 31 | 0.02 | 53 | 56 |
| | | | | | | | | |
| H | Aug 2019 | 0.95 | 0.85 | 0.87 | 17 | 0.01 | 44 | 67 |
| | Sep 2019 | 0.95 | 0.9 | 0.92 | 10 | 0.74 | 44 | 65 |
| | | | | | | | | |
| I | Aug 2019 | 1.78 | 0.95 | 0.98 | 17 | 0.03 | 55 | 20 |
| | Sep 2019 | 1.72 | 1.21 | 1.29 | 14 | 0.02 | 52 | 27 |

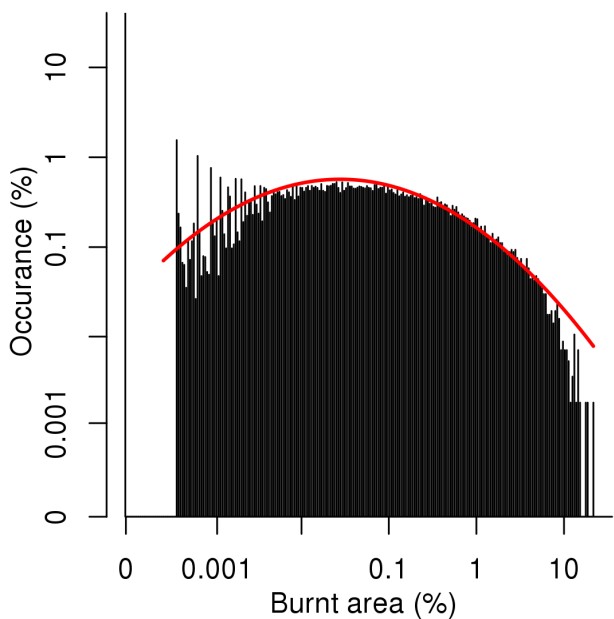

**Figure A1: Distribution of burnt areas in MODIS Collection 6 MCD64A1 burned area product (Giglio et al., 2018) and (red line) fitted normal distribution of logit transformed burnt areas greater than 0.**

600

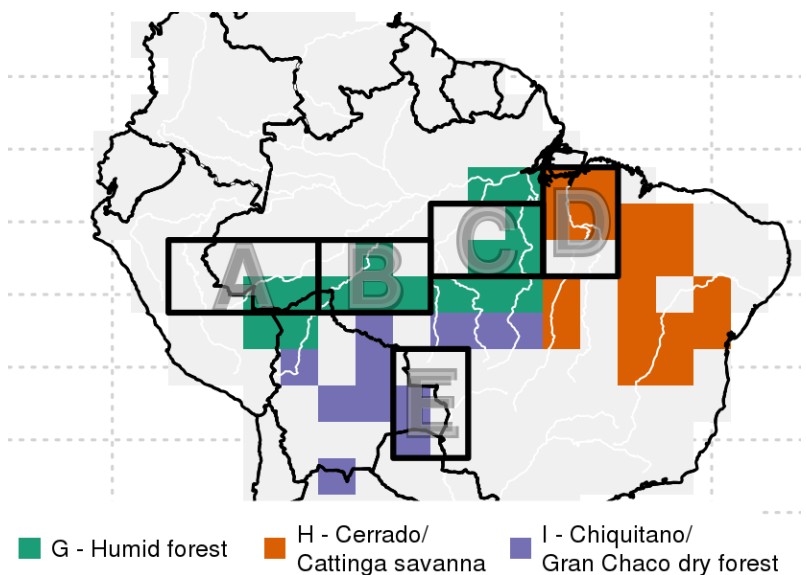

**Figure A2: Study regions. Boxes mark areas used for time series in Fig 1 and rows A-E in Table 1. Coloured areas for time series if Fig 2 and F-I in Table 1, with the entire coloured region being used for F AAD. See Fig. S4 for construction of AAD and areas G-I.**

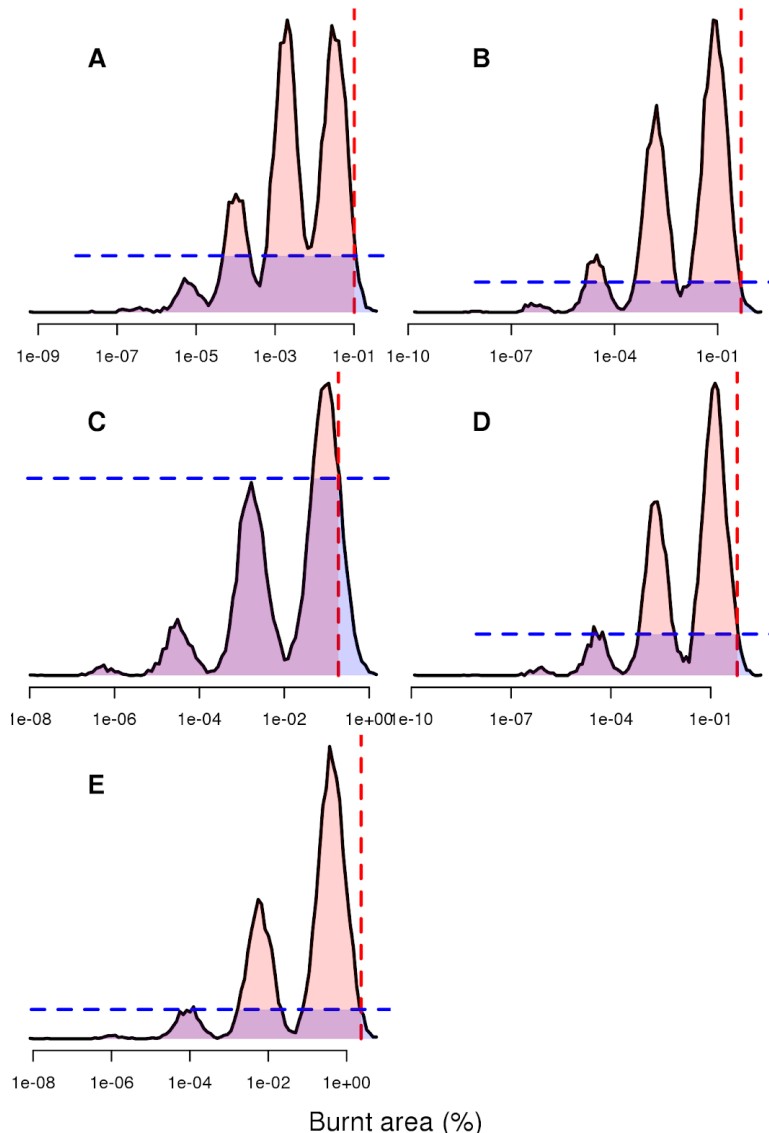

Burnt area (%)

**Figure B1:** Full model posterior solution (black line) for August 2005 across each of the sub-regions compared to MODIS Collection 6 MCD64A1 burned area product (Giglio et al., 2018) (red dashed line). Red shaded area (posterior solution smaller than observed) shows the likelihood high burnt areas were influenced by factors external to the modelling framework. Blue shaded area is the area of the posterior which has less chance of occurrence than the observed burnt area (given by blue dashed line).

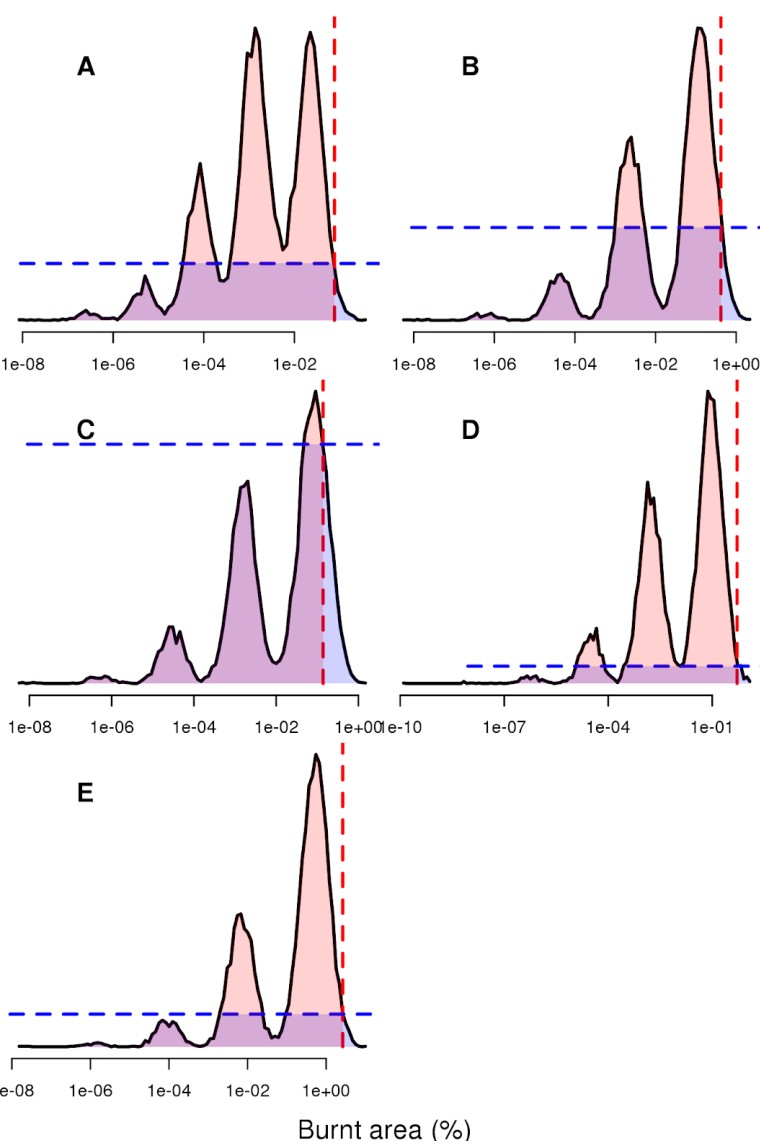

**Figure B2:** As Fig. B1 but for August 2010.

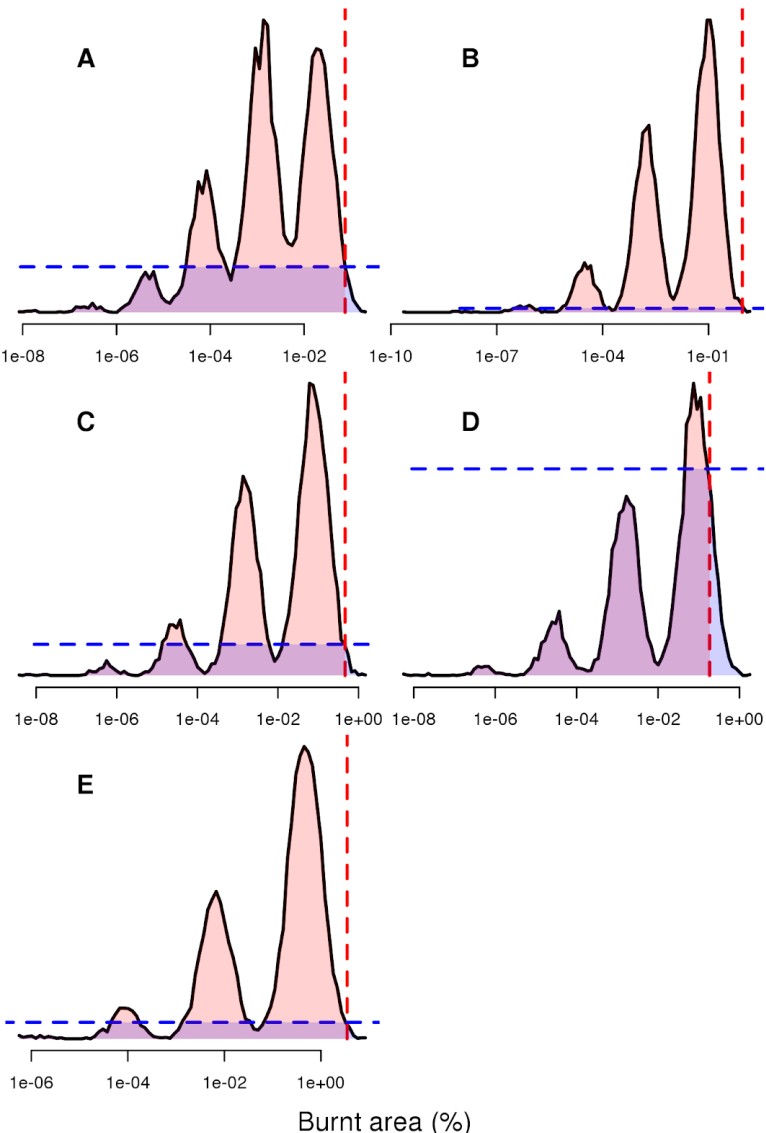

**Figure B3:** As Fig. B1 but for August 2019.

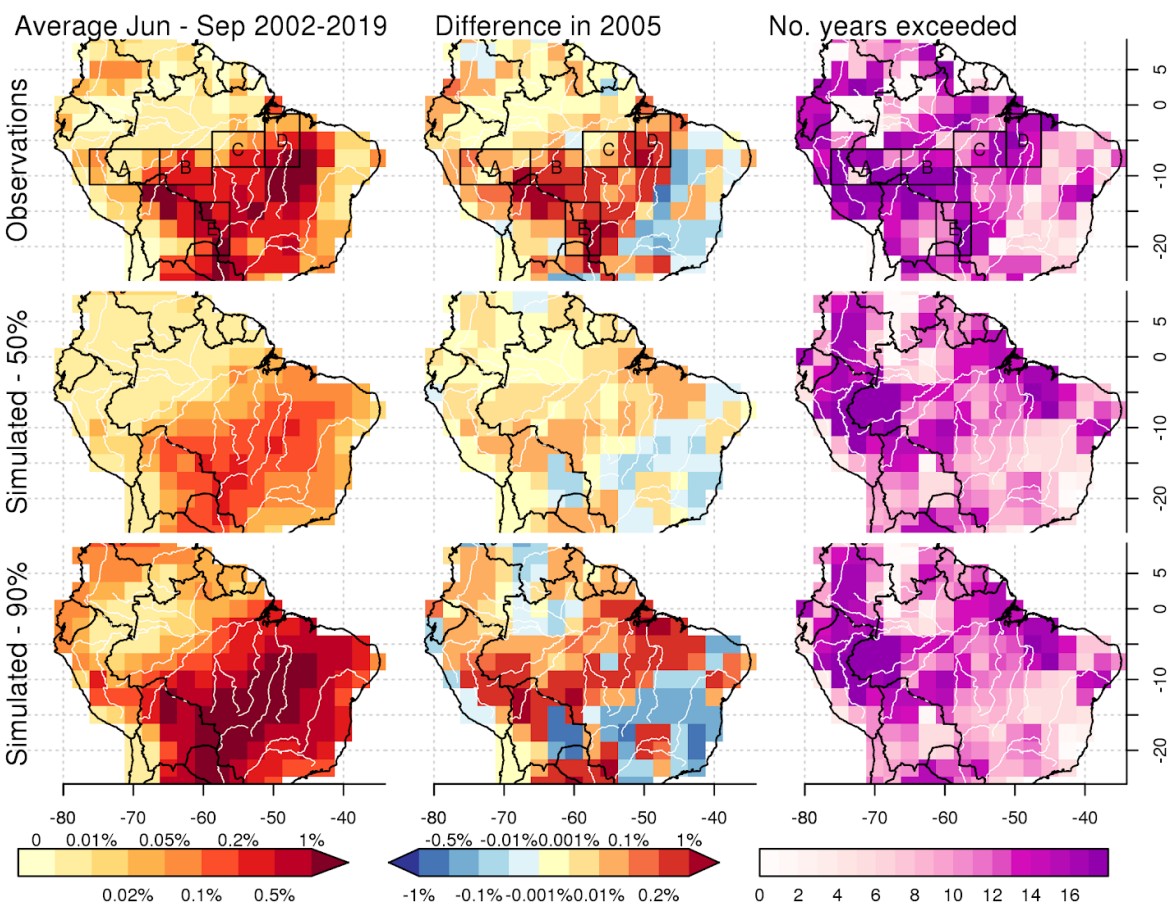

 **Figure C1: Same as Fig. 3 but for June-September annual average compared to 2005.**

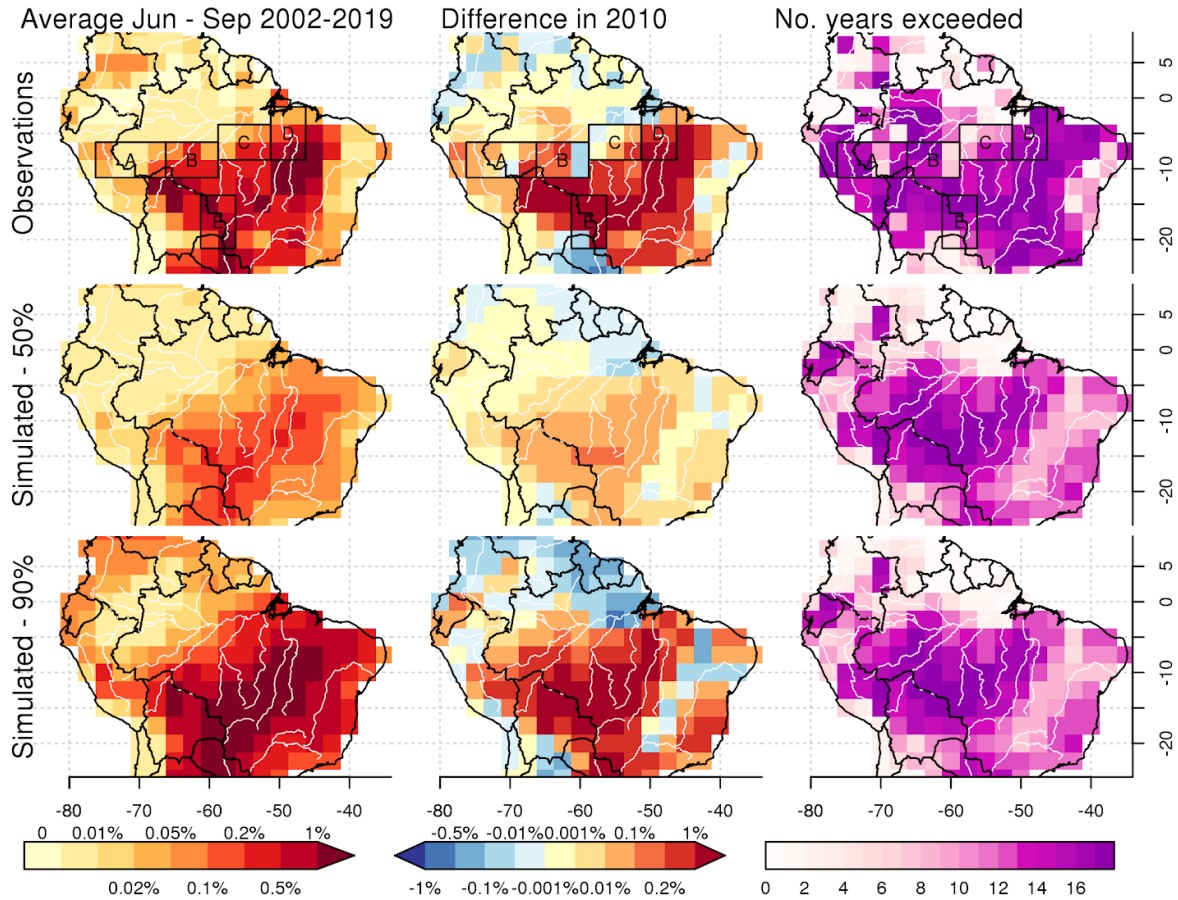

**Figure C2: Same as Fig. C1 but for 2010.**