# Peer review of "Technical note: Low meteorological influence found in 2019 Amazonia fires"

_Biogeosciences, 2020_

## Referee Comment (RC1) · Anonymous Referee #1 · 11 Jun 2020

This manuscript investigates the probability that the burned area anomaly in Amazonia in 2019 was caused by anomalous meteorological conditions. The presented approach is interesting and allows a quick evaluation of recent fire events. They estimate the probability that the event is caused by meteorological conditions using a model setup that includes the meteorological forcing, but not the changes in ignitions caused by humans. This is theoretically a valid approach, however, it remains unclear to me how well the model is able to capture extremes caused by meteorological conditions. I recommend to improve the discussion of the model performance by investigating the model performance for the investigated regions specifically for years that are known to have anomalies in observed fire occurrence caused by meteorological conditions. Source code and data are made available as recommended in best practice guidelines. The

manuscript is well written. The presentation requires some clarifications as indicated in the specific comments.

Specific comments:

p.1 l. 27: Last sentence of abstract should be a conclusion rather than another result.

P2. l. 65: what exactly is a loose attribution? I guess it comes down to interpreting correlations, which can easily be confounded due to the many drivers, as causations?

P3 l.71: actually FDIs often make assumptions on the available fuel, for instance the difference in fuel drying between 1hr, 10hr, 100hr and 1000hr fuels in the fire danger index used in SPITFIRE. Isn't this the basis for Kelley and Harrison 2014 as well?

p.3 l. 81, you could add Forkel et al. 2017 (GMD)

p.3 l. 77: I believe the main disadvantage of fire models embedded in vegetation models is the complexity of the whole model that makes it difficult to fuse the model quickly with most recent observations. Also the inputs from the vegetation model are limiting the model performance. Having a simple fire parameterization which is largely driven by observations clearly has an advantage here. But can not represent the feedbacks between vegetation and fire on the other hand or estimate impacts on the carbon cycle, hydrology etc.

p. 3 l.85: again I think the main advantage is that it is largely driven by observations and it is optimized using observations.

p. 3 l. 85: track uncertainty in the model? Uncertainty in the model suggests that you may refer to the uncertainty related to the model structure, e.g. the shape of functions you implement and the choice of drivers. Usually such bayesian frameworks capture uncertainties of the optimized parameters and can propagate these uncertainties to the output variables. (which is also possible with other optimization techniques.) Please be more precise.

L .109: something wrong with the sentence. Which variable has such coarse resolution? You might reconsider interpolating that variable to higher resolution and therefore being able to maintain the information of the subgrid heterogeneity of other variables could be advantegeous.

l. 155 based on what information did you choose these areas?

l. 165: maybe possible to describe the technique briefly to not make the reader go back to Kelley again?

l. 168: unclear, please rephrase

l. 174: so 1 and 3 are basically the same? 3 for annual 1 for monthly burned area?

l. 179: how do you define areas of recent deforestation? Please indicate these areas.

l. 217: the authors write that in no region the observed anomaly has been that far outside the model range as in 2019. If I look at figure 1 I get a different impression: Region C: 2019 within full posterior of the model, 2007,2010, 2012, 2017 not. Region d: 2010 and 2004 seem much more outside the model range than 2019.

l. 246: what is the novelty in your bayesian approach?

l. 249: explain the setup of your model: it underpredicts burned area when taking into account meteorological conditions but keeping land use and population density constant.

l. 445: Figure 1 caption, please explain the color of the columns on the left.

---

## Referee Comment (RC2) · Sönke Zaehle (Referee) · 21 Aug 2020

Kelly et al. apply a simple numerical model to assess whether or not the 2019 fire season in the Amazonas region was cause by climatic anomalies or other causes. Based on an ensemble of simulations they suggest that the increase in areas of high deforestation was unlikely to be the consequence of climate anomalies and are more likely related to forestry activities. The manuscript is well written, clearly structured and the documentation guidelines for code are followed.

I agree with reviewer #1 that while this is a valid and interesting approach, some discussion should be devoted to the ability of the model to robustly capture the observed interannual variability and specifically the extremes.

[Figure]

Minor comments:

P1L16: Can you be specific here based on what type of information the model is optimised?

P1L18: This makes it sound as if the model redicted an increase in burnt area in these regions, whereas as far as I understand this is actually based on observations. Please clarify.

P3L74ff: Please see comment by reviewer #1.

P3L85: As far as I understand the set-up, the method tracks uncertainty in the parameter set of the model, but does not address alternative model structures or uncertainty in observations. Please be more precise here.

P4L132: Please provide a motivation for this change in approach, and clearify that you mean to say you included all data points in the assimilation procedure?

P5L144: Please used evaluated or similar in stead of validated. Please provide a succint description of the evaluation result here so ease of reading the paper.

P6L168: Hard to follow, please rephrase

P6L174: How is this different from 1. Please also check language of the sentence "Calculated as". What has been calculated?

P6L217: Visually, this isn't true for all the areas and years, can you please provide a quantitative assessment to back up this point?

P8L246: "Novel" makes this read is if the model had been developed in this paper, whereas indeed you have "simply" (correctly and usefully) applied a model published in 2019. Please rephrase and emphasise the novel aspects.

Fig. 2 In order to have the paper readable without the SI, would it be possible to add the regions A-F and ADD to the Figure 2?

---

## Author Comment (AC1) · 7 Oct 2020

Thank you to reviewer #1 for a positive and very useful review. The suggestion that we should explore model performance in previous dry years has led to more extensive revisions then we suspect the reviewer anticipated, and we have outlined the resultant change to the model in a separate author comment. Here, we have made detailed responses to all these major points raised, specific comments and technical corrections, below each comment. We also outline revisions made to the manuscript where appropriate. Reviewer comments are in normal text, responses are in blue, and text quoted from the main text or SI are in *italics* or, for extensive quotes, indented in Times New Roman. Line numbers quoted below are for the original text unless otherwise stated.

This manuscript investigates the probability that the burned area anomaly in Amazonia in 2019 was caused by anomalous meteorological conditions. The presented approach is interesting and allows a quick evaluation of recent fire events. They estimate the probability that the event is caused by meteorological conditions using a model setup that includes the meteorological forcing, but not the changes in ignitions caused by humans. This is theoretically a valid approach, however, it remains unclear to me how well the model is able to capture extremes caused by meteorological conditions. I recommend to improve the discussion of the model performance by investigating the model performance for the investigated regions specifically for years that are known to have anomalies in observed fire occurrence caused by meteorological conditions.

In addition to the changes outlined in our separate author comment on changes in the model's error term description, we have included four new appendix figures (Fig. B1, B2, C1, C2, see figures at the end of this response) that demonstrate the model is able to capture levels of burning 2005 and 2010, years previously already highlighted (as discussed in the main text, lines 185-191) as driven by drier conditions. We have also included a new section 3.1 which describes burnt areas over our observational period, and the model's ability to capture inter-annual variability in burnt areas:

**3.1 Burnt area from 2001-2018**

[revised manuscript text omitted]

Figure C1: Same as Fig. 3 but for 2005.

[Figure]

**Figure C1: Same as Fig. 3 but for 2010.**

Source code and data are made available as recommended in best practice guidelines. The manuscript is well written. The presentation requires some clarifications as indicated in the specific comments.

**Specific comments:**

p.1 l. 27: Last sentence of abstract should be a conclusion rather than another result.
Yes, this is a discussion point of the paper, which is why we have included in the abstract. We have included a result in that abstract that backs this statement up. The sentence now reads "*Burnt area for September in the arc of deforestation had a 31% probability of being caused by meteorological conditions, potentially coinciding with a shift in fire-related policy from South American governments.*"

P2. l. 65: what exactly is a loose attribution? I guess it comes down to interpreting correlations, which can easily be confounded due to the many drivers, as causations?
Yes, that's exactly right, and we have adapted the sentence so it now reads "*However, these do not consider the complex interaction of multiple drivers on fire and are therefore unable to go*

*beyond a loose attribution of a particular forcing to fire, which can easily be confused due to the many drivers, as causations."* to make this point.

P3 l.71: actually FDIs often make assumptions on the available fuel, for instance the difference in fuel drying between 1hr, 10hr, 100hr and 1000hr fuels in the fire danger index used in SPITFIRE. Isn't this the basis for Kelley and Harrison 2014 as well?

Models such as SPITFIRE do apply an FDI to describe the moisture content and drying of different fuel classes, but not the amount of fuel, and rely on some form of vegetation modelling to supply information on the abundance of fuel, which is the point we are making here. We have adapted the sentence to make it clear that FDIs do sometimes form part of vegetation-fire models. *"...such as burnt area or number of fires.* **Some LSM fire schemes achieve this by modelling fuel moisture using FDIs** *(Lenihan et al., 1998; Rabin et al., 2017; Venevsky et al., 2002)".*

It should be noted that the SPITFIRE variant in (Kelley and Harrison, 2014) does not use an FDI. In this model, the Nesterov FDI used in most other SPITFIRE based models was replaced by a more mechanistic description of fuel moisture based on equilibrium moisture content in relation to atmospheric drying potential driven by relative humidity, temperature and timing of rainfall events (see (Kelley et al., 2014) for more details). Additionally, (Kelley and Harrison, 2014) specifically makes the point that changes in FDI are not appropriate for assessing changes in observed fire measures such as burnt area because they lack information from other controls. We, therefore, feel it is appropriate to keep this reference.

p.3 l. 81, you could add Forkel et al. 2017 (GMD)
Included as suggested.

p.3 l. 77: I believe the main disadvantage of fire models embedded in vegetation models is the complexity of the whole model that makes it difficult to fuse the model quickly with most recent observations. Also the inputs from the vegetation model are limiting the model performance. Having a simple fire parameterization which is largely driven by observations clearly has an advantage here. But can not represent the feedbacks between vegetation and fire on the other hand or estimate impacts on the carbon cycle, hydrology etc.
This is an excellent point, and we have added a sentence on the problems of running rapid, near-time assessments at the end of this paragraph (line 81):

> Embedding fire within a complex vegetation model system also prevents rapid observation-model fusion, as iterative optimization techniques are too computationally expensive, instabilities arise from non-linear responses of fire to simulated vegetation and fuel dynamic. Many large scale vegetation-modelling projects, therefore, simulate up to a "present-day" that can be several months or years out of date (Friedlingstein et al., 2019; Hantson et al., 2020).

We had made the point that vegetation-fire models were designed for a different purpose (carbon, hydrology etc) in a previous sentence. (line 74)

p. 3 l.85: again I think the main advantage is that it is largely driven by observations and it is optimized using observations.
We agree, and therefore now start the sentence on with *"**The main advantage** of this system is that it can assess…"*

p. 3 l. 85: track uncertainty in the model? Uncertainty in the model suggests that you may refer to the uncertainty related to the model structure, e.g. the shape of functions you implement and the choice of drivers. Usually such bayesian frameworks capture uncertainties of the optimized parameters and can propagate these uncertainties to the output variables. (which is also possible with other optimization techniques.) Please be more precise.
We have clarified that we mean model parameters by changing the sentence to read *"The main advantage of this system is that it can assess the contribution of different fire drivers directly from observations and track uncertainty in the model parameters and the model's ability to reproduce observations."*. Although *"model's ability to reproduce observations."* is technically represented by two error terms that are included in the modelling framework, we separate out the performance terms as they are not part of the model that represents a physical process. They are also important in translating the more traditional style model output into a useful model posterior that can be used for attributing specific months.

L .109: something wrong with the sentence. Which variable has such coarse resolution? You might reconsider interpolating that variable to higher resolution and therefore being able to maintain the information of the subgrid heterogeneity of other variables could be advantegeous.
It was only land-use data and burnt area that was on a higher-resolution grid. As burnt area was not used to drive the model, we decided that the advantages of higher-resolution for just a small set of drivers were not worth the computation expense of optimising the model at the higher resolution grid. We have now made this clear by replacing the first sentence in the paragraph starting line 109 to:

> All variables were resampled and, where necessary, interpolated to a monthly time-step as per Kelley et al. (2019). All driving variables were provided on a resolution of 2.5° except land use, provided at 0.5°. We, therefore, choose to regrid all datasets to a resolution of 2.5°, as interpolating to a finer resolution would provide no new information about the meteorological drivers tested.

l. 155 based on what information did you choose these areas?
In the original version of the m/s, these were pragmatic choices of areas of definite mismatch. With the new error term of the model, we have been able to focus in on specific areas and transects of historic deforestation activity, and have made this clear by replacing lines 154-165 with:

> We chose five regions (marked A-E in Fig. 1, 2. See Fig. A2 for locations) to represent forest areas already under pressure from deforestation. Regions A-C form a transect (west to east)  across the

agricultural-humid tropical forest interface in Brazil's arc of deforestation, often associated with deforestation (Fig. S4), whereas D and E regions are found in agricultural regions embedded in savanna and grassland regions that experience regular fires:

A. Acre, Southern Amazonas States and Brazilian/Peruvian border
B. Rondônia and Northern Mato Grosso, Brazil.
C. Tocantins, Brazil,
D. Maranhão and Piau in coastal deforestation regions
E. Brazilian, Bolivian and Paraguayan border

We also assessed an overall Area of Active Deforestation (AAD) in the Amazon region (Fig. A2). This area is defined as the parts of South American southern tropics with significant decreasing tree cover trends, as seen in VCF (Dimiceli et al., 2015) and increasing agricultural fractions in the HYDEv3.1 dataset (Klein Goldewijk et al., 2010). Trend analysis used the same technique described in Kelley et al. (2019), where we took significance as $p < 0.05$ on the linear trend for each month in the year on logit transformed variables, using the greenbrown R package (Forkel and Wutzler, 2015). AAD was additionally assessed over 3 sub-regions areas, primarily to evaluate the models' historic performance and assess the increase in 2019 fires across the humidity gradient:

F. Area of Active Deforestation
G. The Southern end of agricultural-**humid tropical forest** interface in Brazil's central states, often associated with arc of deforestation in Brazil's central states
H. **Drier** savanna and woodland in Cerrado and Caatinga in the eastern Basin were land has already been heavily converted to agriculture
I. Southern **dry-deciduous** Chiquitano and Gran Chaco forests, mainly along the Amazon, La Plata watersheds.

l. 165: maybe possible to describe the technique briefly to not make the reader go back to Kelley again?
We have included a brief description of the trend analysis and, perhaps more importantly, acknowledged the software used to perform it by adding the sentence this sentence (see the response to the last comment): "Trend analysis used the same technique described in Kelley et al. (2019), where we took significance as p<0.05 on the linear trend for each month in the year on logit transformed variables, using the greenbrown R package (Forkel and Wutzler, 2015)."

l. 168: unclear, please rephrase
We have tried to clarify each measure with a brief description in bold, and have also provided much more detail as to how each measure was constructed. Lines 168-175 have been replaced with:

1. The **likelihood of observed monthly burnt area** based on the information provided to our model. In the predictive model, the probability of a burnt area $y$ (where $y$ can be outside training data $Y_s$, as is the case for our year 2019 analysis) being explained by our model ($Pred(y)$ - full model uncertainty, or model error, in tan areas on time series in Fig. 1, 2) is proportional to the probability of $y$ given a parameter set, $\beta$, weighted by $P(\beta|Y_s)$ :

$$Pred(y) \ \propto \ \int_{PS} P(\beta|Y_s) \ \times P(y|\beta) \ d\beta \tag{5}$$

Where the observed burnt area, $y$ falls within the model's full posterior ($L(y)$) is then the sum of all probabilities greater than $y$,

$$L(y) \ = \ \int_y^1 Pred(y) \ dBA \tag{6}$$

As our posterior solution is not normally distributed, observations can fall at the extremes of the posterior (i.e when there is no burnt area, $y = 0$ and by definition $D(y) \ = \ 0$), and still have a high likelihood (i.e if $P(Y_s = 0|\beta)$ in equation (3) is much greater than 0. See Fig B2 as an example). We, therefore, define the significance of $D(y)$ as the probability of $y$ occurring by chance ($pv(y)$) from the sum of all probabilities below $P(y)$ (Fig. A2):

$$pv(y) \ = \ 1 \ - \ \sum_{i \varepsilon 2Z}^{i \le n} \left[ \int_{y_{i-1}}^{y_i} P(x) \ dx \ - \ P(y) \ \times (y_i \ - \ y_{i-1}) \right] \tag{7}$$

where $\{y_1, \ ...., \ y_n\}$ is the set of solutions to $P(y_i) \ = \ P(y)$

Whenever $D(y)$ and $pv(y)$ is low indicates burning significantly higher than expected than suggested by the model in that month.

2. The **likelihood burnt area would have been higher than the annual average**, i.e the fraction of the model's full posterior greater than the model's annual average climatological posterior (the point where the vertical lines cross 1 in right-hand columns in Fig. 1,2). A climatological burnt area *clim* for a given month, $m$, in the year (i.e January, February, etc) can be calculated from the convolution of each year's posterior solutions, $\beta_{yr, m}$. Not that it's the model inputs, incorporated in $\beta_m$, and not the model parameters that vary with time:

$$P(BA \mid clim_m) \ = \ P\left(BA \mid \beta_{2001, m} \ \cap \ \beta_{2002, m} \ \cdots \ \cap \ \beta_{2019, m}\right)$$

Where $P\left(BA/2 \mid \beta_i \cap \beta_j\right) = \int_0^{BA} P\left(BA - x \mid \beta_j\right) \times P\left(BA \mid \beta_i\right) dx$ and

$$P\left(BA/3 \mid \beta_i \cap \beta_j \cap \beta_k\right) = \int_0^{BA} P\left(BA - x \mid \beta_k\right) \times P\left(x/2 \mid \beta_i \cap \beta_j\right) dx \qquad (8)$$

The probability of an anomaly $A$ in a given year, $yr$, for month $m$ is, therefore:

$$P\left(A \mid \beta_{yr,\,m} \cap clim_m\right) = \int_0^1 P(A \times BA \mid \beta_{yr,\,m}) \times P(BA \mid clim)\, dBA \qquad (9)$$

The likelihood of a year having a higher anomalous, $A$, is the sum of probabilities of $A < 1$:

$$L\left(A \mid \beta_{yr,\,m} \cap clim_m\right) = \int_0^A P\left(A \mid \beta_{yr,\,m} \cap clim_m\right) dA \qquad (10)$$

And the likelihood of the year an average burnt area is given $L\left(1 \mid \beta_{yr,\,m} \cap clim_m\right)$

3. The **likelihood of the observed anomalous year** occurring is given by :

$$L\left(\frac{y_{m,\,yr}}{\overline{y_m}} \,/\mid \beta_{yr,\,m} \cap clim_m\right) \qquad (11)$$

where $y_{m,\,yr}$ is the burnt area for the month, m, and year, yr, in question, and $\overline{y_m}$ is the climatological average of that month

l. 174: so 1 and 3 are basically the same? 3 for annual 1 for monthly burned area?
This should now be clearer in the revised text, outlined above,  3 is roughly equivalent to 1, but on climatological anomaly rather than raw values.

 l. 179: how do you define areas of recent deforestation? Please indicate these areas.
Areas of decline are areas showing a decreasing trend in MODIS VCF tree cover (Dimiceli et al., 2015) and increasing agriculture in HYDE (Klein Goldewijk et al., 2010) (see the response to comment on line 165). The areas were shown in Supplementary Fig. S4, but we have also included a map of regions as a new appendix Fig A2, which we refer to at appropriate points in the text.

[Figure]

**Figure A2:** Study regions. Boxes mark areas used for time series in Fig 1 and rows A-E in Table 1. Coloured areas for time series if Fig 2 and F-I in Table 1, with the entire coloured region being used for F AAD. See Fig. S4 for construction of AAD and areas G-I.

l. 217: the authors write that in no region the observed anomaly has been that far outside the model range as in 2019. If I look at figure 1 I get a different impression: Region C: 2019 within full posterior of the model, 2007,2010, 2012, 2017 not. Region d: 2010 and 2004 seem much more outside the model range than 2019.

The "anomaly" is in reference to the scatter plots on the right of Figure 1, and not the time series. We have added a brief discussion on the months where observations fall outside the time series posterior in section 3.1 (see the response to the reviewer's main comment), and have made it clear, whenever appropriate in the main text, that anomaly refers to the scatter plots in Fig 1 and (new) Fig. 2 (see additional author comment 3). The discussion on observed vs modelled anomalous years at the end of (now) section 3.3, " *Climatic conditions in 2019*" reads:

> The observed anomaly for August 2019 is higher than the model across all regions except. This is particularly prominent in regions B and C where observations show that burnt area was 196% and 138% greater than the August average (Table 1). Whereas the model suggests that meteorological conditions alone should have resulted in a fire season with a 16-122% (based on 5-95% parameter uncertainty range for parameter uncertainty) increase in burnt area in B and 2% reduction to 4% increase for C compared to the August average, with only a 57% and 53% chance of a greater burnt area than the average for B and C respectively. The likely occurrence of the observed anomaly was 7 and 10% for B and C respectively (Table 1) - much greater than any previous year (Fig 1B and C, August column).

> The higher observed anomaly vs the model extends over much of the AAD (Fig. 2 "August" column, red points). The model suggests a 4-6% reduction for the AAD, with a 49% probability of greater than the annual average burnt area (Table 1). By comparison, the observed burnt area was 45% greater than the annual average, with a 20% likelihood. Again, the observed anomaly seems to be least likely in more humid regions. For our humid area, G, the model suggests a small (10-14%) increase in burnt area, with only a 12% probability of 107% increase seen in observations, whereas the 5% observed reduction in drier savanna regions in H seems to be in line with the model (at 30% likelihood).

l. 246: what is the novelty in your bayesian approach?
See the response to reviewer #2

l. 249: explain the setup of your model: it underpredicts burned area when taking into account meteorological conditions but keeping land use and population density constant.

We have added *"... when taking into account meteorological conditions but keeping land use, population density and human-fire interactions constant"* to this sentence, which now reads:

> The model predicts a lower burnt area than we see in the observations for Amazonia during June-August
> 2019, **when taking into account meteorological conditions but keeping land use, population density**

**and human-fire interactions constant. This indicates** that from observed meteorological data alone, we would not expect 2019 to be a high-fire year.

l. 445: Figure 1 caption, please explain the color of the columns on the left.
We have added *"The colour indicates the year, with 2019 in red."* to the caption.

---

## Author Comment (AC2) · 8 Oct 2020

Thank you for this positive review. As with reviewer 1, the suggestion that we should explore model performance in previous years has led to extensive revisions, outlined in more detail in a separate author comment. Here, we have made detailed responses to all these major points raised, specific comments and technical corrections, below each comment. We also outline revisions made to the manuscript where appropriate. Reviewer comments are in normal text, responses are in blue, and text quoted from the main text or SI are in *italics* or, for extensive quotes, indented in Times New Roman. Line numbers quoted below are for the original text unless otherwise stated.

Kelly et al. apply a simple numerical model to assess whether or not the 2019 fire season in the Amazonas region was cause by climatic anomalies or other causes. Based on an ensemble of simulations they suggest that the increase in areas of high deforestation was unlikely to be the consequence of climate anomalies and are more likely related to forestry activities. The manuscript is well written, clearly structured and the documentation guidelines for code are followed. I agree with reviewer #1 that while this is a valid and interesting approach, some discussion should be devoted to the ability of the model to robustly capture the observed interannual variability and specifically the extremes.

In addition to the appendix figures included in response to reviewer 1, we have also incorporated into the new section 3.1 a description of historic annual, seasonal and inter-annual burning in tropical South America, and discussed how well our model reproduces these fire regime measures. This includes dry years and years of increased human fire activity. We show the model captures years of increased burning driven by meteorological conditions and, as expected given the model set up, the increased uncertainty associated with human driven-year season fires and years of increased deforestation. See response to reviewer 1 for changes.

**Minor comments:**

P1L16: Can you be specific here based on what type of information the model is optimised?
We have adapted the sentence (line 16) to read *"To answer this, we take advantage of a recently developed modelling framework which optimises a simple fire model against observations of burnt area, and whose outputs are described as probability densities."*

P1L18: This makes it sound as if the model redicted an increase in burnt area in these regions, whereas as far as I understand this is actually based on observations. Please clarify.
We replace *"We show…" at the start of this sentence* with *"Observations show…"* to  clarify.

P3L74ff: Please see comment by reviewer #1.
Please see the response to reviewer #1

P3L85: As far as I understand the set-up, the method tracks uncertainty in the parameter set of the model, but does not address alternative model structures or uncertainty in observations. We
We have clarified this in response to reviewer #1's second comment on line 85

P4L132: Please provide a motivation for this change in approach, and clearify that you mean to say you included all data points in the assimilation procedure?

Kelley et al. 2019, a global study on a 0.5-degree grid, and made the pragmatic choice of sampling 10% of the observations due to computational demands. We did not have the same demands in this study because of our coarser resolution and sample study area. We have now made this clear in the text. We also rephrased "data point" with "grid cells" to match terminology typically used for this type of fire modelling and made it clear that we are talking about the assimilation procedure. The sentence now reads *"We used all of the 44750 grid cells on our 2.5° grid and monthly time step for 16 years **in our assimilation procedure, This is a departure from Kelley et al.** (2019)**, where only 10% of grid cells were used, as our sample size was much smaller and we did not face the same computational demand."***

P5L144: Please used evaluated or similar in stead of validated. Please provide a succint description of the evaluation result here so ease of reading the paper. P6L168: Hard to follow, please rephrase

We have replaced "validated" with "evaluated" and provide a little bit more detail of the benchmarking methods implemented. By adding the following sentence to line 144:

> … parameter uncertainty of our model, corresponding to the yellow areas in time series in Fig. 1. The mean burnt area for a particular parameter combination ($\overline{BA_\beta}$), was obtained from:
>
> $$\overline{BA_\beta} = \int_0^1 P(BA|\beta) \times BA \ dBA \qquad (4)$$
>
> $\overline{BA_\beta}$ was evaluated using the implementation of the fireMIP benchmarking metrics (Rabin et al. 2017; Hantson et al. 2020) as per Kelley et al. (2019).

A summary of evaluation results is now provided in the new section 3.1, as outlined in the response from both reviewers regarding model performance.

P6L174: How is this different from 1. Please also check language of the sentence "Calculated as". What has been calculated?

We have expanded our description of the 3 measures. See response to reviewer #1s comment in line 174 of the original text.

P6L217: Visually, this isn't true for all the areas and years, can you please provide a quantitative assessment to back up this point?

With clarification of the three different measures of likelihood, and the change in figure caption in response to a similar comment by reviewer 1, we hope it is now clear we refer to the scatter points on the left of figure 1, and not the time series.

P8L246: "Novel" makes this read is if the model had been developed in this paper, whereas indeed you have "simply" (correctly and usefully) applied a model published in 2019. Please rephrase and emphasise the novel aspects.

While the model has been developed in this paper (especially in the revised version of the m/s), it is the "approach" that is novel, not the model itself. By applying a Bayesian framework, we are able to the test observations as opposed to the model. For Bayesian statisticians, we foresee this to be particularly exciting as it mirrors the inverse framing of the question associated with Bayes theorem. I.e what is the chance of the observations given the modelling framework. That might  be to technical for a conclusion in an interdisciplinary journal, so we have chosen to simply add *"... by testing the likelihood of observed extremes in fire again inferred historic relations*" to this sentence (line 246)

 Fig. 2 In order to have the paper readable without the SI, would it be possible to add the regions A-F and ADD to the Figure 2?
A-F have been added to observed maps of fire in Figs 3,4, C1 and C2. We have included a map of AAD as Fig. A2 (see reponse to reviewer #1s comment online 179).

Hantson, Stijn, Douglas I. Kelley, Almut Arneth, Sandy P. Harrison, Sally Archibald, Dominique Bachelet, Matthew Forrest, et al. 2020. "Quantitative Assessment of Fire and Vegetation Properties in Historical Simulations with Fire-Enabled Vegetation Models from the Fire Model Intercomparison Project." *Geoscientific Model Development Discussions*. https://doi.org/10.5194/gmd-2019-261.
Kelley, Douglas I., Ioannis Bistinas, Rhys Whitley, Chantelle Burton, Toby R. Marthews, and Ning Dong. 2019. "How Contemporary Bioclimatic and Human Controls Change Global Fire Regimes." *Nature Climate Change* 9 (9): 690–96.
Rabin, Sam S., Joe R. Melton, Gitta Lasslop, Dominique Bachelet, Matthew Forrest, Stijn Hantson, Fang Li, et al. 2017. "The Fire Modeling Intercomparison Project (FireMIP), Phase 1: Experimental and Analytical Protocols." *Geoscientific Model Development* 20: 1175–97.

---

## Author Comment (AC3) · 8 Oct 2020

We would like to thank both reviewers for their positive and useful comments on our m/s. The suggestion that we evaluate model performance in previous years turned out to be particularly valuable. It demonstrated that the assumption of normally distributed model errors (equation 3 in the original m/s), along with a small (and now corrected) error in processing lightning inputs, made it difficult for the model to capture previous high fires years within it's posterior accurately. We felt that the ability of the model to reproduce the historical record, particularly in dry years, was a pre-requisite to applying the framework to assess meteorological influences over 2019 fires. As such, we have implemented a new error term to improve the model performance in previous extreme years. The most substantial revisions of the m/s with regard to this specific change are outlined below.

In the revised m/s, we now demonstrated that a logit-transformed, zero-inflated normal distribution is a much more appropriate way of representing the conditional probability of observation given parameter combinations, by including a new appendix Fig A1:

[Figure]

**Figure A1: Distribution of burnt areas in MODIS Collection 6 MCD64A1 burned area product (Giglio et al., 2018) and (red line) fitted normal distribution for logit transformed burnt areas greater than 0.**

We have also replaced our error description and the probability of observations given a parameter combination (lines 126-129) with:

> 41.47% of the burnt area observations are zero, and the remaining are normally distributed under *logit* transformation (Fig. A1). We, therefore, defined the likelihood, $P(Y_s|\beta)$, using a zero-inflated normal distribution on the logit transformed burnt area, as opposed to a simple normal distribution as used in Kelley et al (2019). This better described the observational to the simulated difference in burnt area during times of very low or very high burning. Our zero-inflation likelihood term is therefore described as:
>
> $$P(Y_s = 0|\beta) = 1 - BA_i^2 \times (1 - P_0)$$
>
> $$P(Y_s > 0|\beta) = [1 - P(Y_s = 0|\beta)] \times \frac{N}{\sigma\sqrt{2\pi}} exp\left\{\Sigma_i^N\left(\frac{logit(y_i) - logit(BA_i)}{\sigma}\right)^2\right\} \qquad (3)$$
>
> where $i$ represents an individual data point, $y_i$ is the burnt area observations, $N$ is the observation sample size and $logit(x) = log\left(\frac{x}{1-x}\right)$.

This new term allows for a much narrower uncertainty range at low burnt areas (see tan in the original and revised Figure 1) and, consequently, allows for a broader error term during periods of extreme burning. The models full posterior now captures levels of burning during historic dry years, as described by the response to reviewer #1s main comment.

The extra detail the model provides means we can focus on capturing drivers over smaller geographical areas. We have therefore modified our regions slightly to capture the west-east transect across the arc of deforestation in Brazil and to explain variations in burning throughout our area of active deforestation (AAD), though the AAD itself remains unchanged. We have therefore changed the region description (lines 154-161) to:

A. Acre, Southern Amazonas States and Brazilian/Peruvian border
B. Rondônia and Northern Mato Grosso, Brazil.
C. Tocantins, Brazil,
D. Maranhão and Piau in coastal deforestation regions
E. Brazilian, Bolivian and Paraguayan border
F. Area of Active Deforestation (AAD)
G. [the areas of the AAD that is on] Southern end of agricultural-**humid tropical forest** interface in Brazil's central states, often associated with arc of deforestation in Brazil's central states
H. [the areas of the AAD that is on] **Drier** savanna and woodland in Cerrado and Caatinga in the eastern Basin were land has already been heavily converted to agriculture
I. [the areas of the AAD that is on] Southern **dry-deciduous** Chiquitano and Gran Chaco forests, mainly along the Amazon, La Plata watersheds.

See the response to reviewer #1 for all changes in relation to region descriptions.

We have split Figure 1 in two (i.e new Fig1. and Fig 2, with Fig. 2 and 3 in the original manuscript becoming 3 and 4). Notice the narrower posterior in tan and model performance compared to the original version

[Figure]

**Figure 1: Time series and fire season anomalies for modelled and observed burnt area. See Fig. A2 for locations of A-R. Red lines show monthly burnt area observations from MCD64A1, yellow shows model accounting for parameter uncertainty (10-90%) and brown shows full model uncertainty (10-90%). The red line is dashed when observations and model accounting for parameter uncertainty overlap. Vertical grid lines are positioned for August each year. Right-hand plots show observed (x-axis) and modelled (y-axis) anomaly, calculated as 2019 burnt area over 2002-2019 climatological average burnt area for (first column) August and (second column) September. The colour indicates the year, with 2019 in red. Thin lines show 10-90% full model uncertainty, while dots and thick line indicate 10-90% parameter uncertainty**

[Figure]

**Figure 2: As Fig. 1, but for the "Area of Active Deforestation" region which incorporates areas where there has been a significant increase in agriculture and decrease in tree cover. See Fig. S4, and regions and increased agriculture and decreased tree cover in the (G) humid tropical forest, (H) savanna and (I) dry-deciduous Forest.**

We also have modified the results and figures to capture these new insights (see attached revised m/s), though the overall conclusion of the paper - that meteorological conditions did not drive the increased burning in deforestation areas of South America in 2019, remain unchanged.

---

## Author Response (AR1)

We have provided a point-by-point description of the changes to the m/s in our responses to reviewers (see AC1 and AC2). Also note our AC3, where we outline changes to the conditional probability of observations given a parameter combination.

On top of the changes outlined in these responses, we have also updated the m/s as follows:
1. Line 47-55 (of the revised m/s) to reflect the rapid changing literature surrounding recent fire events
2. We have modified section 3.2 (line 273 - 284) to avoid repetition with the new section 3.1
3. In addition to the new section 3.1 outlined in our response to reviewer 1, we have adapted the results to reflect changes in the model. This includes the 2nd half of the abstract and a new part of section 3.3, lines 311-321 where we separate a description of anomaly comparisons.
4. We updated benchmark comparisons and figures in the SI.
5. Corrected English and made grammatical improvements throughout

Below is a tracked changed version of the main paper and SI,

Please let us know if any more information is required.

Thanks

Douglas Kelley on behalf of co-authors

[revised manuscript text omitted]

---

## Author Response (AR2)

Thank you for accepting the manuscript. Note that in the uploaded text, there is a small number of corrected spelling mistakes, an inclusion of a reference to coastlines used in the maps, and the code and data availability links now point to the latest versions used to produce the accepted results.

While the title still starts with "Technical note:", we were wondering if this is still appropriate given the length of the post-reviewed manuscript. We will be happy with any editorial decision in this regard.

Thank you

Douglas Kelley, on behalf of co-authors.